# A Unified Framework for Deep Hypergraph Clustering Beyond Homophily

Bowen Zhao [1]   Qianqian Wang [1]

## Abstract

Deep hypergraph clustering exhibits compelling capacity for node representation learning via modeling high-order relationships. However, most existing methods adopt fixed propagation mechanisms and implicitly assume homophily, which presumes that adjacent nodes possess similar characteristics. This assumption might deviate from real-world situations, particularly under heterophilic conditions, thereby degrading clustering performance. To address this limitation, we propose a **Uni**fied Framework for **D**eep **H**ypergraph **C**lustering (Uni-DHC). Specifically, we design a learnable high-order hypergraph propagation strategy that fuses multi-order information and adaptively learns their importance derived from raw data. To stabilize unsupervised training and eliminate structural redundancy caused by high-order aggregation, we additionally enforce node-level consistency and hyperedge-level decorrelation constraints. From the spectral perspective, we demonstrate that conventional HGNN-style propagation corresponds to a fixed low-pass filter, whereas our designed method induces a learnable polynomial spectral filter. Extensive experiments on homophilic and heterophilic datasets illustrate that Uni-DHC consistently outperforms state-of-the-art methods, achieving prominent performance improvement in heterophilic settings.

## 1. Introduction

Hypergraphs extend conventional graphs by enabling a single hyperedge to connect multiple nodes simultaneously. Owing to this property, hypergraphs can model high-order relationships that cannot be represented by pairwise edges (Bretto, 2013). This merit renders hypergraphs suitable for diverse practical scenarios, including social networks (Li

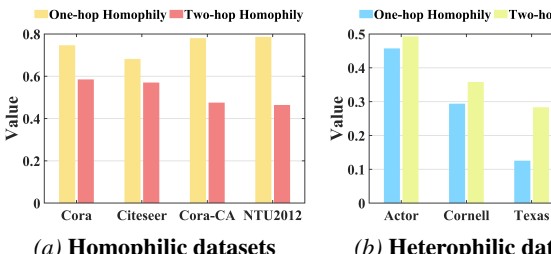

*Figure 1.* 1-hop and 2-hop homophily ratio on representative datasets. The results show that homophily varies across propagation ranges, which motivates adaptive high-order propagation.

et al., 2013; Zhu et al., 2018), biological systems (Feng et al., 2021; Dai & Gao, 2023), and recommendation systems (Xia et al., 2021; Yu et al., 2021), where interactions often involve groups of entities rather than isolated pairs.

Motivated by the success of graph neural networks (GNNs) (Scarselli et al., 2008; Chen et al., 2024), numerous hypergraph neural networks (HNNs) (Feng et al., 2019; Cai et al., 2022) have been developed to extract latent representations from hypergraph topological structures. These methods generally adopt a message passing scheme, in which node features are first aggregated into hyperedges and then propagated back to nodes. Most existing HNNs are designed for supervised or semi-supervised tasks, *e.g.,* node classification. Nevertheless, annotating hypergraph data is expensive and time-consuming. Consequently, massive hypergraph data remain unlabeled, which restricts practical deployment of supervised HNNs. To tackle the problem, hypergraph clustering is explored as a feasible unsupervised method.

Hypergraph clustering aims to partition nodes into disjoint groups without label guidance. Early efforts primarily extend spectral clustering to hypergraph (Zhou et al., 2006; Yang et al., 2021). Despite their simplicity in implementation, such methods fail to model complex high-order structural patterns. Recently, deep hypergraph clustering methods (Wang et al., 2025b) based on contrastive learning have attained superior performance, which generally construct multiple augmented views for hypergraphs and learn node representations by maximizing inter-view consistency.

Nevertheless, most deep hypergraph clustering techniques adopt static propagation strategies with 1-hop message passing. These approaches implicitly build on the homophily

---

[1]School of Telecommunications Engineering, Xidian University, Xi'an, Shaanxi, China. Correspondence to: Qianqian Wang <qqwang@xidian.edu.cn>.

*Proceedings of the 43rd International Conference on Machine Learning*, Seoul, South Korea. PMLR 306, 2026. Copyright 2026 by the author(s).

assumption, *i.e.*, nodes linked via hyperedges share similar features. Under this premise, iterative propagation smooths node features and enhances representations. However, it does not always hold in practice, as hyperedge-linked nodes may fall into distinct semantic groups. Hence, static 1-hop propagation mixes dissimilar features and degrades clustering performance. Some methods adopt 2-hop propagation to address heterophily but might further suffer performance loss under different homophily levels. Overall, static propagation lacks adaptability across diverse scenarios.

To better illustrate this issue, we investigate homophily across different propagation ranges on four homophilic datasets and four heterophilic datasets. As shown in Figure 1a, homophilic datasets typically present strong 1-hop homophily, whereas their 2-hop homophily varies across datasets and can differ noticeably from the 1-hop case. In contrast, Figure 1b indicates that heterophilic datasets yield much weaker 1-hop homophily, whereas their 2-hop homophily reveals distinct structural patterns. These observations suggest that meaningful information is not confined to a fixed local neighborhood. Therefore, rigidly adopting a fixed propagation depth may overlook informative high-order patterns or introduce noise, which motivates the need for adaptive high-order propagation.

In this paper, we propose a unified framework for deep hypergraph clustering beyond homophily (Uni-DHC). The core of Uni-DHC is a learnable high-order hypergraph propagation module. Instead of fixing the propagation depth, our method aggregates information across multiple propagation orders and learns their importance end-to-end from data. This enables the model to flexibly balance local and long-range structural information. Furthermore, Uni-DHC incorporates two self-supervised objectives to enhance representation quality. At the node level, a cross-view alignment module enforces consistency between different semantic projections. At the hyperedge level, we adopt a correlation reduction module to eliminate redundant information in hyperedge representations arising from node membership overlaps. These two components constitute an effective framework for unsupervised hypergraph clustering. We further provide a theoretical analysis demonstrating that conventional HGNN-style propagation corresponds to a fixed low-pass spectral filter. In contrast, our learnable propagation can be considered as a polynomial spectral filter with adaptive frequency responses. The main contributions of this paper are summarized as follows:

- We propose Uni-DHC, a unified framework for deep hypergraph clustering that alleviates the challenges posed by heterophily in hypergraph structures.

- We introduce a node-level cross-view alignment and a hyperedge-level correlation reduction objective, which

jointly enhances representation consistency and mitigates structural redundancy in an unsupervised way.

- We conduct a spectral analysis to illustrate that conventional HGNN-style propagation corresponds to a fixed low-pass filter, whereas our propagation scheme achieves adaptive spectral responses.

## 2. Related Work

Motivated by the powerful representation capability of contrastive learning, recent works have developed self-supervised objectives tailored for hypergraph representation learning. HyperGCL (Wei et al., 2022) constructs augmented views and leverages contrastive learning to enhance representation quality under low-label settings. TriCL (Lee & Shin, 2023) adopts a tri-directional contrastive strategy that jointly models node, hyperedge, and node–hyperedge membership relations, capturing both node-level and group-level structural information. MMACL (Lee & Chae, 2024) integrates high-order and pairwise relationships via multi-view mixed attention and leverages contrastive learning to learn expressive node representations for hypergraphs. CHGNN (Song et al., 2024) enhances hypergraph representations by jointly learning adaptive views and contrastive objectives over hypergraph structures. SE-HSSL (Li et al., 2024) improves hypergraph self-supervised learning by leveraging correlation-based objectives to reduce reliance on negative sampling. HyFi (Roh et al., 2024) refines hypergraph contrastive learning by defining weak positive pairs and using noise-based augmentation to capture high-order commonalities and meanwhile preserves the original structure. HCN (Wang et al., 2025b) simplifies hypergraph clustering by replacing convolution with smoothing preprocessing and enforcing consistency and structural alignment. Despite their promising performance, most methods implicitly favors homophilic structures. Consequently, they tend to degrade discriminative features in heterophilic hypergraphs, where adjacent nodes often exhibit distinct properties.

## 3. Method

In this section, we present a unified framework for deep hypergraph clustering beyond homophily (Uni-DHC). As shown in Figure 2, the framework comprises three key modules, *i.e.*, learnable high-order hypergraph propagation, node-level cross-view alignment, and hyperedge-level correlation reduction module. In the reminder of the section, we systematically illustrate each module and explain their contributions to the entire framework.

### 3.1. Preliminaries

**Notions:** Denoting the hypergraph as $\mathcal{H} = \{\mathbf{X}, \mathbf{H}, \mathcal{E}\}$, we define $\mathcal{V} = \{v_1, v_2, \ldots, v_n\}$ as a set of $n$ nodes associated

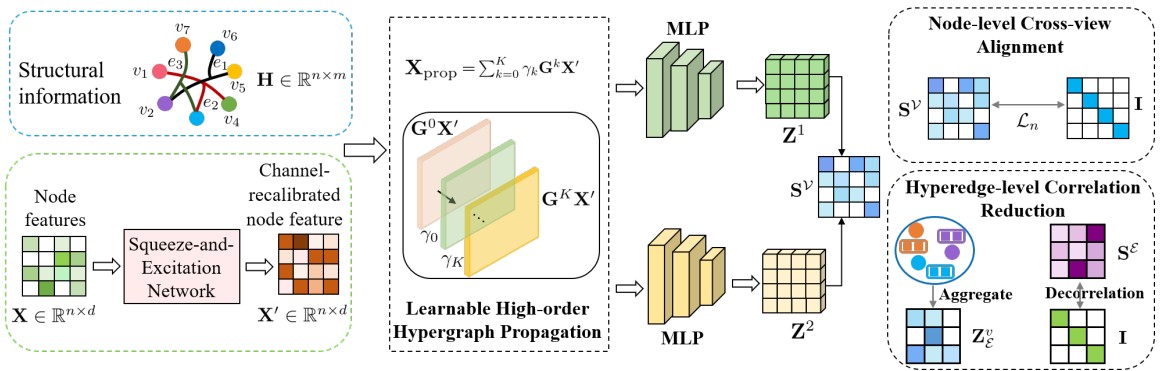

*Figure 2.* Illustration of the overall framework of the proposed Uni-DHC. Specifically, SE-based channel recalibration refines node features, which are then aggregated by a learnable weighted mixture of multi-step hypergraph propagations to obtain high-order representations; two unshared MLP encoders produce dual-view embeddings, regularized by node-level cross-view alignment and hyperedge-level correlation reduction, and the fused embedding is finally used for clustering.

with $\mathcal{C}$ classes, and $\mathcal{E} = \{e_1, e_2, \ldots, e_m\}$ as a collection of $m$ hyperedges. The node feature matrix is represented by $\mathbf{X} \in \mathbb{R}^{n \times d}$, where $d$ is the dimensionality of node features. The incidence matrix $\mathbf{H} \in \{0, 1\}^{n \times m}$ characterizes the membership relations between nodes and hyperedges, such that $\mathbf{H}_{ij} = 1$ if node $v_i$ participates in hyperedge $e_j$, and $\mathbf{H}_{ij} = 0$ otherwise. Furthermore, we denote $\mathbf{D}_v \in \mathbb{R}^{n \times n}$ as the diagonal matrix of node degrees and $\mathbf{D}_{\mathcal{E}} \in \mathbb{R}^{m \times m}$ as the diagonal matrix of hyperedge degrees.

**Problem Statement:** Deep hypergraph clustering aims to learn meaningful node representations in an unsupervised manner and group them into distinct clusters. Specifically, a hypergraph neural network $\mathcal{F}$ is employed to capture node features $\mathbf{X}$ and structural information encoded in $\mathbf{H}$, producing low-dimensional embeddings $\mathbf{Z}$. Then, it performs clustering such as k-means or spectral clustering on representations to partition the nodes into $\mathcal{C}$ disjoint clusters.

**Definition 3.1 (High-order Neighborhoods in Hypergraphs).** Given a hypergraph $\mathcal{H} = \{\mathcal{V}, \mathcal{E}\}$, a path of length $k$ connecting nodes $u$ and $v$ is defined as a sequence of $k$ hyperedges where consecutive hyperedges share common nodes. The shortest path distance between $u$ and $v$ is denoted by $d_{\mathcal{H}}(u, v)$. We define the 1-hop neighborhood of $v_i$ as the set of nodes that share at least one common hyperedge with $v_i$, namely $\mathcal{N}_1(v_i) = \{v_j \in \mathcal{V} \setminus v_i \mid \exists e \in \mathcal{E}, v_i, v_j \subseteq e\}$. To capture long-range structural dependencies beyond local connectivity, for $k \geq 2$ we further define the strict $k$-hop neighborhood as the set of nodes that are exactly $k$ steps away from $v_i$, that is, $\mathcal{N}_k(v_i) = \{v_j \in \mathcal{V} \mid d_{\mathcal{H}}(v_i, v_j) = k\}$.

**Definition 3.2 (Hypergraph Homophily).** Given a hypergraph $\mathcal{H} = \{\mathcal{V}, \mathcal{E}\}$, we first induce a weighted graph by normalized clique expansion:

$$A_{ij} = \sum_{e \in \mathcal{E}, \{v_i, v_j\} \subseteq e} \frac{1}{|e| - 1}. \quad (1)$$

Based on the induced graph, 1-hop homophily is computed as the weighted ratio of same-label node pairs among directly connected nodes:

$$\mathcal{H}_1 = \frac{\sum_{i \neq j} A_{ij} \mathbb{I}(y_i = y_j)}{\sum_{i \neq j} A_{ij}}. \quad (2)$$

Similarly, 2-hop homophily is measured over strict 2-hop relations obtained from $A^2$ after removing self-loops and direct 1-hop connections.

**Definition 3.3 (Hypergraph-induced Propagation Operator).** We first define node affinity matrix $\mathbf{M} = \mathbf{H} \mathbf{D}_{\mathcal{E}}^{-1} \mathbf{H}^{\top}$. Based on $\mathbf{M}$, the hypergraph-induced propagation operator is defined as $\mathbf{G} = \mathbf{D}^{-\frac{1}{2}}(\mathbf{M} + \mathbf{I})\mathbf{D}^{-\frac{1}{2}}$, where $\mathbf{D}$ is the diagonal degree matrix of $\mathbf{M} + \mathbf{I}$. Intuitively, $\mathbf{M}$ encodes the strength of pairwise relationships between nodes by aggregating their co-occurrences within shared hyperedges. The operator $\mathbf{G}$ then normalizes these relationships into a propagation operator, enabling stable and efficient aggregation of high-order structural information in the hypergraph.

### 3.2. Learnable High-order Hypergraph Propagation

To enhance the expressiveness of node features prior to hypergraph propagation, we introduce a channel-wise feature recalibration mechanism based on the Squeeze-and-Excitation Network (Hu et al., 2018). Given a node feature matrix $\mathbf{X} \in \mathbb{R}^{n \times d}$, we first aggregate global statistics across all nodes by average pooling along the node dimension:

$$s_c = \frac{1}{n} \sum_{i=1}^{n} X_{i,c}, \quad c = 1, 2, \ldots, d. \quad (3)$$

The resulting channel descriptors are then processed by a lightweight gating network consisting of two fully connected layers with ReLU and Sigmoid activations, producing a set of channel-wise attention weights:

$$\mathbf{u} = \sigma\big(\delta(\mathbf{s}\mathbf{W}_1)\mathbf{W}_2\big), \quad (4)$$

where $\mathbf{W}_1 \in \mathbb{R}^{d \times d/r}$ and $\mathbf{W}_2 \in \mathbb{R}^{d/r \times d}$ are learnable parameters, $\delta(\cdot)$ denotes the ReLU function, $\sigma(\cdot)$ is the Sigmoid function, and $r$ controls the reduction ratio. Finally, the original features are recalibrated by channel-wise scaling with the learned attention weights, $\mathbf{X}' = \mathbf{X} \odot \mathbf{u}$ where $\mathbf{X}' \in \mathbb{R}^{n \times d}$ denotes the channel-recalibrated node feature matrix and $\mathbf{u}$ is broadcast along the sample dimension. This adaptive reweighting allows the model to highlight informative feature channels while suppressing redundant ones, leading to a more compact and discriminative representation.

After channel-wise feature refinement, we propagate the enhanced node representations using a hypergraph-induced propagation operator $\mathbf{G}$ to model high-order structural dependencies beyond pairwise interactions. Rather than fixing the propagation depth or restricting aggregation to a single-hop neighborhood, we introduce a learnable high-order propagation scheme that explicitly accounts for contributions from multiple propagation steps. The propagated node representations are computed as:

$$\mathbf{X}_{\text{prop}} = \sum_{k=0}^{K} \gamma_k \mathbf{G}^k \mathbf{X}', \tag{5}$$

where $\mathbf{G}^k$ denotes the $k$-step hypergraph propagation operator and $\gamma_k$ are learnable coefficients controlling the relative importance of different propagation orders.

This formulation allows the model to flexibly balance information from local and more distant neighborhoods. In contrast to conventional hypergraph neural networks that rely on a fixed number of propagation layers, the learnable weights $\gamma_k$ enable the model to emphasize informative propagation ranges while mitigating noisy or misleading structural information. This flexibility is particularly important for hypergraphs with heterophily or complex high-order interaction patterns.

### 3.3. Node-level Cross-view Alignment

After obtaining the propagated node features $\mathbf{X}_{\text{prop}}$, we further project them into multiple view-specific embedding spaces to facilitate cross-view alignment at the node level. Specifically, we employ view-dependent MLP encoders to generate normalized node representations for each view:

$$\begin{aligned}
\mathbf{Z}^1 &= \text{MLP}_1(\mathbf{X}_{\text{prop}}), \quad \mathbf{Z}^1 = \frac{\mathbf{Z}^1}{\|\mathbf{Z}^1\|_2}, \\
\mathbf{Z}^2 &= \text{MLP}_2(\mathbf{X}_{\text{prop}}), \quad \mathbf{Z}^2 = \frac{\mathbf{Z}^2}{\|\mathbf{Z}^2\|_2},
\end{aligned} \tag{6}$$

Although $\text{MLP}_1$ and $\text{MLP}_2$ share the same network architecture, their parameters are not shared, allowing the two views to capture complementary semantic information

during training. We then compute a cross-view similarity matrix between $\mathbf{Z}^1$ and $\mathbf{Z}^2$ as:

$$\mathbf{S}^{\mathcal{V}} = \mathbf{Z}^1 \cdot (\mathbf{Z}^2)^\top \tag{7}$$

where $\mathbf{S}^{\mathcal{V}} \in \mathbb{R}^{n \times n}$ is a cosine similarity matrix, and $\mathbf{S}_{ij}^{\mathcal{V}}$ measures the similarity between the $i$-th node in the first view and the $j$-th node in the second view. To enforce consistency of node representations across different views, we align the similarity matrix with the identity matrix, encouraging each node to be most similar to its counterpart across views. The corresponding alignment loss is defined as:

$$\mathcal{L}_n = \frac{1}{n} \sum_{i=1}^{n} (\mathbf{S}_{ii}^{\mathcal{V}} - 1)^2, \tag{8}$$

Minimizing $\mathcal{L}_n$ promotes cross-view invariance at the node level, preserving shared semantic information while reducing the influence of view-specific noise.

### 3.4. Hyperedge-level Correlation Reduction

While node-level alignment enforces consistency across different views, it does not explicitly address redundancy at the hyperedge level. In hypergraphs, hyperedges often exhibit strong correlations due to overlapping node memberships. This issue becomes more pronounced under high-order hypergraph propagation, where multi-step aggregation repeatedly mixes information from shared nodes, leading to highly correlated and redundant hyperedge representations, which may degrade discriminability.

To alleviate this issue, we introduce a hyperedge-level correlation reduction objective that encourages diversity among hyperedge representations. Given the node representations from two views and the hypergraph incidence matrix, we first construct hyperedge representations by aggregating the embeddings of nodes incident to each hyperedge. Specifically, we apply a pooling operation over nodes belonging to the same hyperedge:

$$\mathbf{Z}_{\mathcal{E}}^v = \mathbf{D}_{\mathcal{E}}^{-1} \mathbf{H}^\top \mathbf{Z}^v, \quad v \in \{1, 2\}, \tag{9}$$

where $\mathbf{D}_{\mathcal{E}}$ denotes the hyperedge degree matrix and mean pooling is adopted by default. The resulting hyperedge representations are then $\ell_2$-normalized and used to compute a cross-view hyperedge similarity matrix:

$$\mathbf{S}^{\mathcal{E}} = \frac{\mathbf{Z}_{\mathcal{E}}^1 \cdot (\mathbf{Z}_{\mathcal{E}}^2)^\top}{\|\mathbf{Z}_{\mathcal{E}}^1\| \|\mathbf{Z}_{\mathcal{E}}^2\|}, \tag{10}$$

To reduce redundancy among hyperedges, we encourage the cross-view similarity matrix $\mathbf{S}^{\mathcal{E}}$ to be close to the identity matrix, thereby promoting high similarity for corresponding hyperedges while suppressing correlations across different

ones.

$$\mathcal{L}_h = \frac{1}{m}\sum_{i=1}^{m}(\mathbf{S}_{ii}^{\mathcal{E}} - 1)^2 + \frac{1}{m(m-1)}\sum_{i \neq j}(\mathbf{S}_{ij}^{\mathcal{E}})^2, \quad (11)$$

By minimizing $\mathcal{L}_h$, the model is encouraged to learn decorrelated and complementary hyperedge representations across views. This hyperedge-level regularization effectively reduces structural redundancy and complements node-level alignment, leading to more expressive and robust high-order representations for hypergraph clustering.

### 3.5. Objective Function

The overall objective of Uni-DHC combines the node-level cross-view alignment loss $\mathcal{L}_n$ and the hyperedge-level correlation reduction loss $\mathcal{L}_h$. In summary, the objective function is defined as follows:

$$\mathcal{L} = \mathcal{L}_n + \beta\mathcal{L}_h, \quad (12)$$

where $\beta$ is a trade-off parameter that balances the contributions of $\mathcal{L}_n$ and $\mathcal{L}_h$. After training, the learned node representations from the two views are fused to obtain a unified embedding:

$$\mathbf{Z} = \frac{1}{2}\left(\mathbf{Z}^1 + \mathbf{Z}^2\right), \quad (13)$$

The fused representation $\mathbf{Z}$ is then used as input to the $k$-means algorithm to cluster nodes into disjoint groups.

## 4. Spectral Interpretation of Hypergraph Propagation

We provide an interpretation from the spectral view for the proposed learnable high-order hypergraph propagation. In particular, we show that it is equivalent to a learnable polynomial filter on the hypergraph spectrum, enabling adaptive use of high-order information beyond homophily.

Let $\mathbf{G} = \mathbf{D}^{-1/2}(\mathbf{M} + \mathbf{I})\mathbf{D}^{-1/2}$, where $\mathbf{M} = \mathbf{H}\mathbf{D}_{\mathcal{E}}^{-1}\mathbf{H}^{\top}$ and $\mathbf{D}$ is the degree matrix of $\mathbf{M} + \mathbf{I}$. Since $\mathbf{M} = \mathbf{H}\mathbf{D}_{\mathcal{E}}^{-1/2}(\mathbf{H}\mathbf{D}_{\mathcal{E}}^{-1/2})^{\top} \succeq \mathbf{0}$, we have $\mathbf{G}$ is symmetric positive semi-definite.[1] Thus, $\mathbf{G}$ admits an eigendecomposition $\mathbf{G} = \mathbf{U}\mathbf{\Lambda}\mathbf{U}^{\top}$.

Our high-order propagation satisfies:

$$\mathbf{X}_{\text{prop}} = \sum_{k=0}^{K}\gamma_k\mathbf{G}^k\mathbf{X}' = \mathbf{U}\left(\sum_{k=0}^{K}\gamma_k\mathbf{\Lambda}^k\right)\mathbf{U}^{\top}\mathbf{X}', \quad (14)$$

which is exactly a polynomial spectral filter $g_{\gamma}(\lambda) = \sum_{k=0}^{K}\gamma_k\lambda^k$ applied to $\mathbf{X}'$.

---

[1]Assuming all degrees are positive so that the normalization is well-defined.

*Table 1.* Statistical characteristics of eight datasets used in our experiments.

| Types | Datasets | Nodes | Hyperedges | Features | Classes |
|---|---|---|---|---|---|
| **Homophilic** | **Cora** | 1,434 | 1,579 | 1,433 | 7 |
| | **Citeseer** | 1,458 | 1,079 | 3,703 | 6 |
| | **Cora-CA** | 2,388 | 1,072 | 1,433 | 7 |
| | **NTU2012** | 2,012 | 2,012 | 100 | 67 |
| **Heterophilic** | **Actor** | 16,255 | 10,164 | 50 | 3 |
| | **Cornell** | 183 | 181 | 1,703 | 5 |
| | **Texas** | 183 | 182 | 1,703 | 5 |
| | **Wisc** | 251 | 248 | 1,703 | 5 |

To connect $\mathbf{\Lambda}$ with frequency, define the normalized Laplacian $\mathbf{\Delta}_G = \mathbf{I} - \mathbf{G}$. As commonly adopted in graph signal processing, eigenvectors associated with small Laplacian eigenvalues correspond to low-frequency (smooth) components. Equivalently, eigenmodes with $\lambda$ close to $1$ in $\mathbf{G}$ are low-frequency, while smaller $\lambda$ indicates relatively higher frequencies.

Unlike fixed propagations with predefined low-pass responses, the coefficients $\{\gamma_k\}$ are learned end-to-end, yielding a data-adaptive spectral response $g_{\gamma}$ that can reweight different frequency bands. This is crucial for heterophilic hypergraphs, where informative patterns may reside in medium/high frequencies that fixed smoothing tends to suppress. Appendix C further shows that HGNN-style propagation corresponds to a fixed low-pass filter under a linearized setting, highlighting the contrast between fixed and learnable propagation.

## 5. Experiment

### 5.1. Experimental Settings

**Datasets.** We conduct the experiments on eight hypergraph datasets, including four homophilic and four heterophilic datasets. Specifically, the homophilic datasets include Cora (Sen et al., 2008), Citeseer (Sen et al., 2008), Cora-CA (Rossi & Ahmed, 2015), and NTU2012 (Chen et al., 2003), while the heterophilic datasets include Actor (Li et al., 2025), Cornell, Texas, and Wisc (Pei et al., 2020). Table 1 summarizes the statistics of these eight datasets.

**Comparing Methods.** To demonstrate the performance of our proposed method, we select nine state-of-the-art methods as comparison methods. These include HyperGCL (Wei et al., 2022), ED-HNN (Wang et al., 2022), TriCL (Lee & Shin, 2023), SheafHyperGNN (Duta et al., 2023), MMACL (Lee & Chae, 2024), CHGNN (Song et al., 2024), SE-HSSL (Li et al., 2024), HyFi (Roh et al., 2024), and HCN (Wang et al., 2025b). Among these, ED-HNN and SheafHyperGNN are improved hypergraph neural networks specifically to handle heterophily, while HyperGCL, TriCL, MMACL, CHGNN, SE-HSSL, HyFi, and HCN are con-

*Table 2.* The results of clustering on homophilic hypergraph datasets. The best and runner-up results are highlighted in **Red** and **Blue**, respectively.

| Dataset | Metric | HyperGCL | ED-HNN | TriCL | SheafHyperGNN | MMACL | CHGNN | SE-HSSL | HyFi | HCN | Uni-DHC |
|---|---|---|---|---|---|---|---|---|---|---|---|
| Cora | ACC | 57.29±3.78 | 23.79±1.78 | 75.46±1.42 | 22.31±1.45 | 73.50±0.61 | 52.63±3.58 | 76.10±0.91 | 75.01±0.68 | **75.95±0.41** | **77.82±0.42** |
| | NMI | 38.51±3.09 | 03.78±0.98 | 57.20±0.98 | 02.84±0.92 | 56.16±0.93 | 41.21±2.35 | **59.01±1.05** | 57.45±1.00 | 58.52±0.59 | **59.61±0.55** |
| | ARI | 31.08±3.93 | 02.13±0.91 | 55.60±1.76 | 01.52±0.72 | 49.63±0.71 | 30.15±3.41 | 53.46±1.74 | 53.56±1.37 | **56.65±0.69** | **57.64±0.97** |
| | F1 | 55.99±3.74 | 22.65±1.61 | 72.69±2.76 | 20.34±1.35 | 69.37±2.33 | 46.33±5.04 | 72.40±1.50 | **73.55±0.96** | 73.54±0.48 | **76.02±0.52** |
| Citeseer | ACC | 49.11±2.54 | 24.46±1.72 | 69.73±0.79 | 23.60±1.33 | 68.78±0.78 | 50.96±2.26 | 68.37±2.04 | 69.25±0.57 | **71.67±0.86** | **72.67±0.99** |
| | NMI | 24.81±3.28 | 02.48±0.94 | 44.27±0.91 | 01.86±0.74 | 44.93±1.06 | 30.78±2.04 | 43.90±1.79 | 44.60±0.77 | **46.85±0.63** | **47.44±0.99** |
| | ARI | 22.85±3.43 | 01.63±0.79 | 45.95±1.34 | 01.07±0.58 | 45.13±0.99 | 28.36±3.03 | 43.41±2.21 | 46.09±1.49 | **47.75±1.26** | **49.31±1.45** |
| | F1 | 45.07±2.56 | 23.58±1.52 | 63.74±1.25 | 22.11±1.48 | 62.25±1.05 | 44.02±1.66 | 62.02±3.03 | 64.37±0.31 | **64.55±1.70** | **65.42±1.87** |
| Cora-CA | ACC | 47.42±4.93 | 22.45±0.72 | 72.04±1.51 | 22.01±1.02 | 71.27±0.94 | 42.50±0.90 | **73.74±1.28** | 69.89±1.04 | 73.59±0.74 | **74.96±0.96** |
| | NMI | 28.66±4.32 | 02.81±0.33 | 51.87±2.44 | 02.19±0.90 | 51.60±1.36 | 29.44±1.02 | **54.06±1.29** | 49.33±1.09 | 53.66±1.20 | **53.76±0.81** |
| | ARI | 21.75±4.83 | 01.58±0.34 | 47.96±2.07 | 01.34±0.61 | 46.43±0.91 | 16.78±4.04 | 49.30±2.12 | 45.13±1.66 | **51.04±1.68** | **52.45±2.09** |
| | F1 | 44.44±4.71 | 21.35±0.65 | 69.28±1.41 | 19.34±1.20 | 67.80±2.00 | 41.74±1.70 | 70.67±2.43 | 68.40±0.94 | **71.68±0.69** | **73.20±0.64** |
| NTU2012 | ACC | 53.71±1.02 | 66.42±1.12 | 69.11±1.33 | 41.43±4.25 | 69.39±1.87 | 61.27±1.55 | 66.58±1.43 | **70.91±0.76** | 70.04±1.26 | **71.32±0.83** |
| | NMI | 72.13±0.41 | 80.56±0.63 | 82.60±0.48 | 60.75±3.15 | 82.66±0.63 | 79.53±0.53 | 81.84±0.55 | **83.28±0.41** | 83.05±0.42 | **83.54±0.41** |
| | ARI | 45.85±1.78 | 61.84±1.77 | 65.03±2.19 | 30.49±5.46 | 65.82±2.89 | 56.05±2.40 | 60.26±2.18 | **68.51±1.25** | 66.61±1.79 | **67.56±1.19** |
| | F1 | 46.69±1.76 | 58.77±1.44 | 61.71±2.06 | 36.66±3.25 | 61.37±1.70 | 53.57±1.88 | 59.33±1.42 | 61.48±1.61 | **62.20±1.45** | **62.36±1.13** |

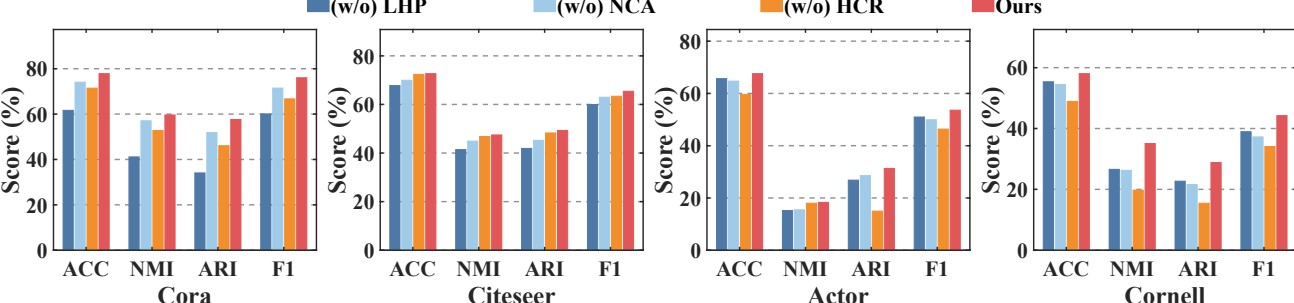

*Figure 3.* Ablation studies on Cora, Citeseer, Actor, and Cornell datasets.

trastive learning-based methods.

**Evaluation Metrics.** Clustering performance is evaluated using four standard metrics: Accuracy (ACC), Normalized Mutual Information (NMI), Adjusted Rand Index (ARI), and Macro F1-score (F1), where higher values indicate better performance. To reduce the effect of randomness, we report the mean and standard deviation of each metric over ten independent runs for every method.

## 5.2. Comparison Experiments

Tables 2 and 3 report the clustering results on homophilic and heterophilic hypergraph datasets. Overall, Uni-DHC achieves best or near-best performance across most metrics, demonstrating its superiority in learning robust and discriminative representations under diverse hypergraph structures.

On homophilic datasets, Uni-DHC exhibits strong adaptability and stability. On widely used datasets such as Cora and Citeseer, where baselines including HyFi and HCN achieve impressive performance, Uni-DHC still delivers marginal yet consistent improvements across all metrics. This indicates that the proposed unified framework does not sacrifice

performance on relatively simple, homophilic structures, and is able to effectively preserve useful local smoothness while incorporating additional high-order information.

More importantly, Uni-DHC achieves performasubstantialnce gains on heterophilic hypergraphs. As shown in Table 3, several baselines suffer from severe performance degradation on challenging datasets such as Texas and Cornell, reflecting their limited ability to handle heterophilic connections and complex high-order interactions. In contrast, Uni-DHC maintains high performance across all heterophilic datasets. For example, on the Texas dataset, Uni-DHC outperforms the suboptimal method (HCN) by approximately 6.7% in Accuracy and 8.6% in NMI. These results demonstrate Uni-DHC effectively captures informative structural patterns beyond homophily, validating the necessity of adaptive high-order propagation and hyperedge-level redundancy reduction in heterophilic settings.

## 5.3. Ablation Study

To verify the effectiveness of each component, we conduct ablation studies by respectively removing LHP, NCA, and

*Table 3.* The results of clustering on heterophilic hypergraph datasets. The best and runner-up results are highlighted in **Red** and **Blue**, respectively. 'OOM' denotes the out-of-memory failure.

| Dataset | Metric | HyperGCL | ED-HNN | TriCL | SheafHyperGNN | MMACL | CHGNN | SE-HSSL | HyFi | HCN | Uni-DHC |
|---|---|---|---|---|---|---|---|---|---|---|---|
| **Actor** | ACC | 42.80±9.91 | 41.25±3.35 | | 46.25±8.10 | 52.70±2.56 | 61.21±0.00 | | **62.29±0.00** | 58.79±4.50 | **67.57±1.44** |
| | NMI | 00.16±0.13 | 02.92±2.22 | OOM | 01.79±1.57 | 11.56±3.42 | 00.05±0.00 | OOM | 00.01±0.00 | **14.76±4.33** | **18.30±2.94** |
| | ARI | 00.43±0.46 | 02.04±1.88 | | 00.72±0.67 | 06.51±2.36 | -0.04±0.00 | | -00.01±0.00 | **14.14±5.11** | **31.30±5.56** |
| | F1 | 31.77±3.25 | 37.42±3.53 | | 33.42±4.33 | 41.56±4.20 | 25.66±0.00 | | 25.59±0.00 | **47.04±4.89** | **53.60±4.09** |
| **Cornell** | ACC | 34.26±1.76 | 35.74±2.52 | 37.38±1.41 | 41.64±2.56 | 41.42±2.38 | 38.74±1.95 | **53.60±1.30** | 45.46±6.50 | 43.22±1.57 | **58.03±2.41** |
| | NMI | 04.43±0.83 | 07.22±1.92 | 08.36±2.52 | 04.48±1.49 | 07.31±1.76 | 08.28±1.12 | **10.16±4.23** | 05.20±0.91 | 07.39±3.95 | **28.20±2.95** |
| | ARI | 02.27±0.99 | 03.14±2.01 | 04.97±2.11 | 02.44±2.20 | 04.84±1.88 | 04.25±2.38 | **12.50±5.61** | 03.74±2.65 | 07.39±3.95 | **28.83±2.52** |
| | F1 | 25.60±1.15 | 26.93±2.90 | 27.79±3.33 | 24.34±1.96 | 26.78±2.13 | 26.44±1.99 | **30.55±3.80** | 23.81±4.24 | 28.20±2.95 | **44.24±3.84** |
| **Texas** | ACC | 38.20±6.22 | 36.61±2.03 | 44.70±1.95 | 45.41±4.00 | 55.41±3.25 | 43.12±2.12 | 54.04±1.95 | 56.18±2.48 | **59.73±0.88** | **66.50±1.66** |
| | NMI | 05.68±1.76 | 06.57±1.79 | 12.91±1.89 | 06.70±2.23 | 19.97±4.16 | 10.72±2.03 | 09.75±3.61 | 14.94±7.18 | **22.91±2.02** | **31.58±2.10** |
| | ARI | 03.86±3.30 | 03.56±1.77 | 12.53±1.71 | 07.49±3.71 | 22.25±4.35 | 08.02±2.24 | 15.63±4.62 | 19.13±12.39 | **28.43±1.57** | **39.53±2.84** |
| | F1 | 25.41±3.51 | 28.28±2.11 | 32.28±1.62 | 25.62±2.15 | 36.97±3.77 | 30.49±1.23 | 28.87±3.14 | 30.38±8.92 | **39.53±2.84** | **46.00±3.25** |
| **Wisc** | ACC | 35.50±4.32 | 38.37±3.43 | 45.66±3.45 | 40.92±2.22 | 48.77±4.00 | 42.79±3.12 | 49.12±2.57 | 44.38±2.76 | **49.68±1.36** | **65.94±2.04** |
| | NMI | 05.62±1.83 | 09.15±2.72 | **22.66±2.94** | 04.58±1.31 | 20.68±1.73 | 13.35±4.81 | 13.84±3.89 | 11.83±4.33 | 16.58±3.51 | **41.21±5.39** |
| | ARI | 03.50±2.47 | 09.15±2.72 | 13.74±2.06 | 03.81±2.35 | 16.79±2.87 | 09.60±4.70 | 10.59±2.99 | 10.53±2.29 | **17.80±2.58** | **46.60±4.36** |
| | F1 | 26.99±5.04 | 30.21±2.64 | 36.13±2.68 | 26.46±1.79 | 36.36±1.53 | 30.91±6.05 | 34.97±4.69 | 31.68±3.09 | **36.67±2.29** | **50.05±3.17** |

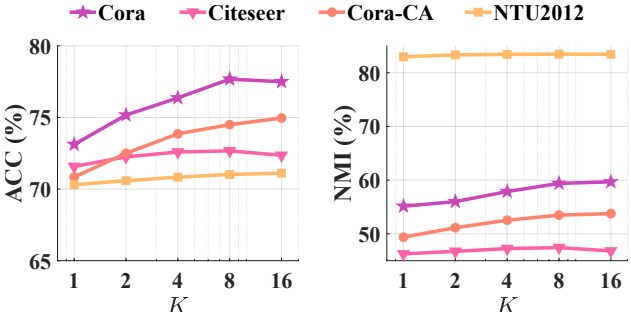

*Figure 4.* The sensitivity analysis of hyper-parameter $K$ across two evaluation metrics on four homophilic datasets.

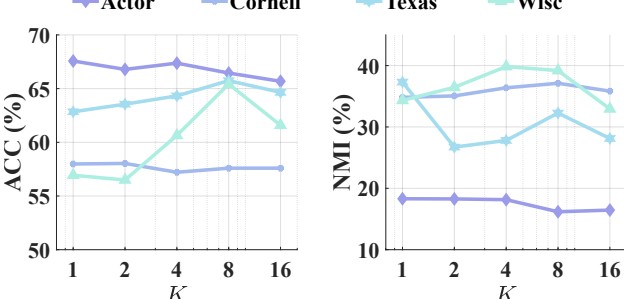

*Figure 5.* The sensitivity analysis of hyper-parameter $K$ across two evaluation metrics on four heterophilic datasets.

HCR. As shown in Figure 3, the full model outperforms all ablated variants, demonstrating the effectiveness of these components. Removing LHP causes clear performance degradation, showing the importance of adaptive high-order propagation. Removing NCA also decreases performance, indicating that node-level alignment helps stabilize representation learning. In addition, excluding HCR leads to notable drops on heterophilic datasets such as Actor and Cornell, confirming the benefit of reducing hyperedge-level redundancy. Appendix Section E provides additional ablations on SE-based channel recalibration and average pooling.

## 5.4. Parameter Sensitivity Analysis

In this subsection, we conduct experiments to analyze the sensitivity of key hyper-parameters in our model. We analyze the sensitivity of two hyper-parameters in our model: the propagation steps $K$ and the trade-off parameter $\beta$. Due to space considerations, the sensitivity results for $\beta$ are reported in Appendix Section E.

**Sensitive analysis of propagation steps $K$.** We evaluate the impact of the propagation step $K$. Specifically, we vary $K$ in the range $\{1, 2, 4, 8, 16\}$. The results are reported in Figures 4 and 5, from which we make the following observations. For homophilic hypergraphs, performance generally improves as $K$ increases and tends to stabilize around $K = 8$. This suggests that capturing long-range, higher-order correlations helps produce more coherent and discriminative node representations for these datasets. In contrast, heterophilic hypergraphs exhibit fluctuating or slightly declining performance when $K$ becomes large. This indicates that excessive propagation may introduce noise and blur discriminative information, and that a relatively smaller receptive field is sufficient to model meaningful local high-order interactions in low-homophily settings.

## 5.5. Time Cost Comparison

We assess the training efficiency of Uni-DHC by comparing its training time with nine baseline methods across eight benchmark datasets. To ensure a fair comparison, all models are trained for 400 epochs under the same experimental

*Table 4.* The training time is compared against nine baseline methods, with all results measured in seconds. "Avg." represents the average training time across the eight datasets.

| Method | Homophilic Hypergraphs | | | | Heterophilic Hypergraphs | | | | |
| --- | --- | --- | --- | --- | --- | --- | --- | --- | --- |
| | Cora | Citeseer | Cora-CA | NTU2012 | Actor | Cornell | Texas | Wisc | Avg. |
| HyperGCL | 34.25 | 29.18 | 38.18 | 88.36 | 560.82 | 7.64 | 6.93 | 7.48 | 96.61 |
| ED-HNN | 2.50 | 2.22 | 2.70 | 11.52 | 3.06 | 1.08 | 1.13 | 1.25 | 3.18 |
| TriCL | 36.56 | 27.72 | 17.12 | 80.56 | OOM | 9.29 | 9.64 | 12.03 | - |
| SheafHyperGNN | 23.14 | 17.85 | 24.31 | 56.25 | 253.61 | 4.58 | 4.65 | 6.66 | 48.88 |
| MMACL | 658.67 | 540.10 | 1048.05 | 1246.02 | 67893.2 | 113.71 | 158.98 | 241.59 | 8987.54 |
| CHGNN | 16.39 | 12.73 | 17.12 | 67.64 | 44.35 | 9.44 | 8.67 | 9.28 | 23.20 |
| SE-HSSL | 54.07 | 30.18 | 82.60 | 13.73 | OOM | 11.98 | 11.82 | 12.89 | - |
| HyFi | 16.42 | 27.37 | 20.38 | 15.53 | 228.84 | 6.67 | 6.23 | 6.62 | 41.01 |
| HCN | 3.36 | 2.56 | 3.71 | 23.19 | 29.61 | 1.34 | 2.26 | 1.51 | 8.44 |
| Ours | **6.75** | **8.03** | **12.19** | **18.70** | **20.46** | **2.95** | **5.39** | **5.19** | **9.96** |

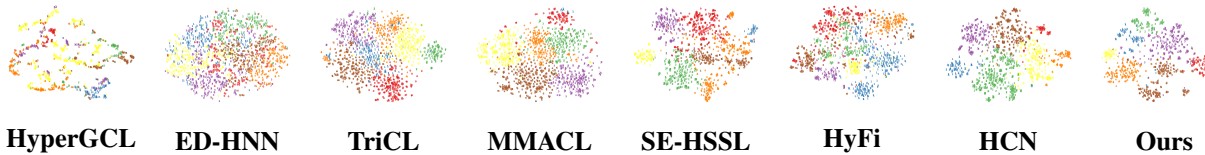

| HyperGCL | ED-HNN | TriCL | MMACL | SE-HSSL | HyFi | HCN | Ours |

*Figure 6.* The visualization of node representation using the t-SNE algorithm on the Cora dataset.

setting. As reported in Table 4, Uni-DHC achieves comparable or faster training speed than most baselines while retaining strong performance. This indicates that our framework attains a favorable balance between effectiveness and efficiency, rather than improving performance at the expense of heavy computational overhead. The efficiency of Uni-DHC mainly stems from two design choices. First, in contrast to many existing hypergraph neural networks that rely on hypergraph convolution, our model employs unshared MLP projections, which substantially reduce the number of trainable parameters. Second, instead of performing computationally intensive hypergraph convolutions, Uni-DHC leverages learnable high-order hypergraph propagation, which streamlines the training process while still capturing rich higher-order structural information.

### 5.6. Visualization Analysis

To qualitatively examine the learned representation space and further validate the effectiveness of Uni-DHC, we adopt t-SNE (van der Maaten & Hinton, 2008) for visualization. In particular, we project the learned node representations on the Cora dataset into a two-dimensional space for comparison. As illustrated in Figure 6, the consensus representations produced by our method exhibit clearly separated and more tightly grouped clusters. This indicates that Uni-DHC can better preserve the underlying semantic structure of the data and capture meaningful latent relationships among nodes, leading to more discriminative and structurally coherent

embeddings than those learned by competing baselines.

## 6. Conclusion

In this work, we propose a unified framework for deep hypergraph clustering beyond homophily, named Uni-DHC, aiming to alleviate the challenges posed by heterophily in hypergraph clustering. To flexibly capture discriminative structural patterns under heterophily, we introduce a learnable high-order hypergraph propagation mechanism that adaptively aggregates information from multiple propagation orders. Instead of relying on fixed-depth propagation, the proposed scheme learns the importance of different ranges directly from data. To stabilize unsupervised training and reduce redundancy introduced by high-order aggregation, we further incorporate node-level cross-view alignment and hyperedge-level correlation reduction objectives. Extensive experiments on both homophilic and heterophilic benchmark datasets demonstrate that Uni-DHC consistently outperforms state-of-the-art hypergraph clustering methods.

## Impact Statement

This paper presents work whose goal is to advance the field of Machine Learning. There are many potential societal consequences of our work, none of which we feel must be specifically highlighted here.

## Acknowledgments

This work is supported by the National Natural Science Foundation of China under Grant 62176203, the Fundamental Research Funds for the Central Universities (ZYTS25267, QTZX25004), and the Science and Technology Project of Xi'an (Grant 2022JH-JSYF-0009), Open Project of Anhui Provincial Key Laboratory of Multimodal Cognitive Computation, Anhui University (No. MMC202416), Selected Support Project for Scientific and Technological Activities of Returned Overseas Chinese Scholars in Shaanxi Province 2023-02.

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

Additional details and results are provided in the appendix. The appendix is organized as follows:

## A. Framework of the Proposed Algorithm

Algorithm 1 summarizes the overall optimization procedure of Uni-DHC. At each training iteration, the model first enhances node features via SE-based channel-wise recalibration, followed by learnable high-order hypergraph propagation to capture structural dependencies. Two view-specific node representations are then generated using unshared MLP encoders, enabling cross-view alignment at the node level. Based on the propagated node representations, hyperedge embeddings are further constructed to reduce redundancy through hyperedge-level correlation regularization. The model is optimized by jointly minimizing the node-level alignment loss and the hyperedge-level correlation reduction loss. After training, the final node representations are obtained by fusing the two views, and standard $k$-means clustering is applied to produce the final clustering results.

---

**Algorithm 1 The optimization of Uni-DHC**

---

**Input**: The input hypergraph $\mathcal{H} = \{\mathbf{X}, \mathbf{H}, \mathcal{E}\}$; The cluster number $\mathcal{C}$; The iteration number $I$; The hypergraph propagation steps $K$; The trade-off parameter $\beta$.

**Output**: The clustering result $\mathbf{Y}$.

1: **for** $i = 1$ to $I$ **do**
2:     Apply SE-based channel-wise feature recalibration to obtain enhanced node features $\mathbf{X}'$ using Eqs. (3)-(4).
3:     Perform learnable high-order hypergraph propagation by Eq. (5).
4:     Generate two view-specific node representations $\mathbf{Z}^1$ and $\mathbf{Z}^2$ using unshared MLP encoders by Eq. (6).
5:     Compute the cross-view node similarity matrix $\mathbf{S}^{\mathcal{V}}$ by Eq. (7).
6:     Calculate the node-level cross-view alignment loss $\mathcal{L}_n$ by Eq. (8).
7:     Aggregate node representations to obtain hyperedge representations $\mathbf{Z}_{\mathcal{E}}^1$ and $\mathbf{Z}_{\mathcal{E}}^2$ by Eq. (9).
8:     Obtain the cross-view hyperedge similarity matrix $\mathbf{S}^{\mathcal{E}}$ by Eq. (10).
9:     Compute the hyperedge-level correlation reduction loss $\mathcal{L}_h$ by Eq. (11).
10:    Optimize the model parameters by minimizing $\mathcal{L}$ by Eq. (12)
11: **end for**
12: Fuse the learned node representations by Eq. (13).
13: Perform $k$-means on $\mathbf{Z}$ to obtain the clustering result $\mathbf{Y}$.
14: **return** $\mathbf{Y}$

---

## B. More Related Work

In recent years, HNNs have achieved remarkable progress by modeling high-order relations that naturally arise in complex systems. Early representative methods, such as HGNN (Feng et al., 2019), extend graph convolutional networks to hypergraphs via hypergraph-induced message passing, enabling effective exploitation of group-wise interactions. Subsequent works further improve the flexibility and expressive power of hypergraph learning. For instance, UniGNN (Huang & Yang, 2021) provides a unified view that generalizes a broad family of GNN architectures to hypergraphs, while AllSet (Chien et al., 2021) leverages multiset functions to design highly expressive hypergraph aggregation operators. Despite these successes, the majority of existing HNNs are primarily developed for supervised tasks (e.g., node classification) and rely heavily on label information for representation learning, which limits their applicability in fully unsupervised settings such as clustering.

Deep hypergraph clustering aims to learn meaningful node representations in a fully unsupervised manner using hypergraph neural networks and to partition nodes into distinct clusters. HCN (Wang et al., 2025b) replaces hypergraph convolution with a preprocessing step and incorporates self-diagonal consistency and structure alignment modules to preserve intrinsic cluster structure while reducing computational cost for hypergraph clustering. HCN-PAI (Wang et al., 2025a) designs a dual-level loss function that enforces consistency at both node and hyperedge levels. Despite their strong performance, most existing methods rely on propagation mechanisms that implicitly favor homophilic structures, which may suppress discriminative signals in heterophilic hypergraphs. Recent efforts have begun to explore hypergraph learning beyond homophily by revisiting propagation designs to better preserve heterophilic information (Li et al., 2025).

## C. Spectral Analysis

In this section, we provide a theoretical justification for the low-pass behavior of HGNN-style propagation. Under a linearized regime, we show that the normalized hypergraph propagation operator induces an implicit low-pass spectral filter, and stacking multiple layers progressively suppresses high-frequency components of hypergraph signals.

**Proposition C.1.** *The HGNN propagation corresponds to an implicit low-pass spectral filter. In particular, stacking $L$ layers of the normalized propagation operator $\mathbf{S}$ progressively attenuates high-frequency components relative to low-frequency structural information.*

*Proof.* Consider a hypergraph $\mathcal{G} = (\mathcal{V}, \mathcal{E})$ with incidence matrix $\mathbf{H} \in \{0,1\}^{|\mathcal{V}| \times |\mathcal{E}|}$. The standard HGNN layer updates node representations by

$$\mathbf{X}^{(l+1)} = \sigma\Big(\mathbf{S}\mathbf{X}^{(l)}\mathbf{\Theta}^{(l)}\Big), \tag{15}$$

where $\mathbf{\Theta}^{(l)}$ is a learnable weight matrix, $\sigma(\cdot)$ is a pointwise nonlinearity, and

$$\mathbf{S} = \mathbf{D}_v^{-1/2}\mathbf{H}\mathbf{W}\mathbf{D}_e^{-1}\mathbf{H}^\top\mathbf{D}_v^{-1/2} \tag{16}$$

is the normalized hypergraph propagation operator. Here $\mathbf{W}$ is a diagonal matrix of nonnegative hyperedge weights, and $\mathbf{D}_v, \mathbf{D}_e$ are diagonal degree matrices of nodes and hyperedges, respectively. We assume all node and hyperedge degrees are positive so that the normalization is well-defined.

To enable spectral analysis, we consider a linearized regime by setting $\sigma(\cdot)$ to the identity and $\mathbf{\Theta}^{(l)} = \mathbf{I}$. Since $\mathbf{W}$ is diagonal with nonnegative entries, it admits $\mathbf{W} = \mathbf{W}^{1/2}\mathbf{W}^{1/2}$. Define:

$$\mathbf{A} = \mathbf{D}_v^{-1/2}\mathbf{H}\mathbf{W}^{1/2}\mathbf{D}_e^{-1/2}. \tag{17}$$

Because $\mathbf{W}$ and $\mathbf{D}_e$ are diagonal (hence commute), we obtain:

$$\mathbf{S} = \mathbf{A}\mathbf{A}^\top, \tag{18}$$

which implies that $\mathbf{S}$ is symmetric positive semi-definite and thus admits an eigendecomposition $\mathbf{S} = \mathbf{U}\mathbf{\Lambda}\mathbf{U}^\top$, where $\mathbf{\Lambda} = \mathrm{diag}(\lambda_1, \ldots, \lambda_n)$ with $\lambda_i \geq 0$.

Next, we show that the spectrum of $\mathbf{S}$ lies in $[0,1]$. Consider the random-walk counterpart

$$\mathbf{P} := \mathbf{D}_v^{-1}\mathbf{H}\mathbf{W}\mathbf{D}_e^{-1}\mathbf{H}^\top. \tag{19}$$

By construction, $\mathbf{P}$ is row-stochastic (its rows sum to one), hence its spectral radius satisfies $\rho(\mathbf{P}) = 1$ and it has an eigenvalue equal to 1. Moreover, $\mathbf{S}$ is similar to $\mathbf{P}$ via $\mathbf{S} = \mathbf{D}_v^{1/2}\mathbf{P}\mathbf{D}_v^{-1/2}$, so $\mathbf{S}$ and $\mathbf{P}$ share the same eigenvalues. Therefore, $\lambda_{\max}(\mathbf{S}) = 1$. Combining this with $\lambda_i \geq 0$ yields $0 \leq \lambda_i \leq 1$ for all $i$.

Define the normalized hypergraph Laplacian $\mathbf{\Delta} := \mathbf{I} - \mathbf{S}$. In hypergraph signal processing, eigenvectors associated with small Laplacian eigenvalues (i.e., $\lambda_i$ close to 1 for $\mathbf{S}$) correspond to low-frequency (smooth) components, whereas large Laplacian eigenvalues (i.e., $\lambda_i$ close to 0 for $\mathbf{S}$) correspond to high-frequency variations.

After stacking $L$ linear propagation layers, the input signal $\mathbf{X}^{(0)}$ becomes

$$\mathbf{X}^{(L)} = \mathbf{S}^L\mathbf{X}^{(0)} = \mathbf{U}\mathbf{\Lambda}^L\mathbf{U}^\top\mathbf{X}^{(0)}. \tag{20}$$

This corresponds to applying a spectral filter with response $g(\lambda) = \lambda^L$. Since $\lambda \in [0, 1]$, $g(\lambda)$ decays rapidly for small $\lambda$, meaning that components associated with $\lambda$ close to 0 (high-frequency modes) are increasingly suppressed as $L$ grows, while components with $\lambda$ close to 1 (low-frequency modes) are relatively preserved. Hence, HGNN-style propagation induces an implicit low-pass filtering effect, and deeper stacking results in stronger smoothing over the hypergraph structure. $\qquad\square$

## D. Experimental Details

### D.1. Heterophilic Hypergraph Datasets

Although HyperUFG (Li et al., 2025) introduces four heterophilic hypergraph datasets, including Actor, Amazon, Twitch, and Pokec, we observe that some of them provide limited discriminative power for evaluating clustering performance. Specifically, Twitch contains only two clusters, and all methods achieve clustering accuracy close to 50%, which is equivalent to random assignment. Therefore, we further construct heterophilic hypergraph datasets from the Cornell, Texas, and Wisc graph benchmarks for evaluating clustering performance. Specifically, for each node $v_i$, we define a hyperedge consisting of $v_i$ and its 1-hop neighbors. This process yields an initial set of hyperedges whose number equals the number of nodes. To ensure meaningful high-order relations, we remove hyperedges whose sizes are smaller than a predefined threshold. In addition, duplicated hyperedges induced by overlapping neighborhoods are eliminated to avoid redundant structural information. The resulting hypergraph thus captures higher-order neighborhood structures while preserving the original heterophilic characteristics of the underlying graph.

### D.2. Implementation Details

All experiments are conducted on a machine equipped with an Intel Core i9-13900K CPU, an NVIDIA GeForce RTX 4090 GPU, and 64 GB of RAM. The models are implemented in PyTorch and trained for up to 400 epochs. We adopt the Adam optimizer to minimize the overall objective, and apply $k$-means clustering on the fused node embeddings to obtain the final results.

## E. Additional Experiment Results

### E.1. Ablation Study

We further analyze how SE-based channel-wise feature recalibration and average pooling in the hyperedge-level correlation reduction module affect clustering performance. The results in Table 5 show that removing the SE mechanism consistently degrades performance across all datasets and evaluation metrics, highlighting the importance of channel-wise feature refinement for learning discriminative representations. We also compare average pooling with alternative aggregation strategies. Overall, average pooling yields more stable and competitive performance than both min pooling and max pooling, especially on heterophilic datasets such as Actor and Cornell. In contrast, min and max pooling tend to amplify noisy or uninformative node features within hyperedges, leading to inferior clustering results. Finally, the complete Uni-DHC model, which combines SE-based feature recalibration with average pooling, achieves the best performance on all four datasets. These results demonstrate that both channel-wise feature refinement and appropriate hyperedge aggregation are critical components for effective hypergraph clustering.

### E.2. Sensitive Analysis of Hyper-parameter $\beta$.

**Sensitive analysis of trade-off hyper-parameter $\beta$.** To further investigate the effect of $\beta$, we perform sensitivity experiments on four datasets by varying $\beta$ in the range $\{0.01, 0.1, 1, 10, 100\}$. The results are shown in Figure 7, from which we draw the following observations. For the homophilic datasets (i.e., Cora and Citeseer), the performance remains relatively stable across different values of $\beta$, indicating that our model is robust to this hyper-parameter on standard citation networks. In contrast, the model is more sensitive to $\beta$ on heterophilic datasets (i.e., Actor and Cornell). In particular, the best performance is achieved at $\beta = 1$, while either excessively small or large values of $\beta$ lead to performance degradation. This suggests that a balanced weighting between node-level and hyperedge-level objectives is crucial for handling heterophilous structures effectively.

*Table 5.* Ablation study on four datasets. The best in each column is shown in **bold**.

| Datasets | Variants | ACC | NMI | ARI | F1 |
|---|---|---|---|---|---|
| Cora | w/o SE | $74.91_{\pm0.85}$ | $57.70_{\pm0.71}$ | $52.86_{\pm1.39}$ | $73.11_{\pm1.52}$ |
| | min pooling | $75.68_{\pm0.77}$ | $58.00_{\pm1.14}$ | $54.42_{\pm1.49}$ | $73.70_{\pm1.08}$ |
| | max pooling | $75.36_{\pm0.73}$ | $57.90_{\pm2.27}$ | $54.34_{\pm1.38}$ | $73.32_{\pm1.27}$ |
| | Uni-DHC | $\mathbf{77.82_{\pm0.42}}$ | $\mathbf{59.61_{\pm0.55}}$ | $\mathbf{57.64_{\pm0.97}}$ | $\mathbf{76.02_{\pm0.52}}$ |
| Citeseer | w/o SE | $71.78_{\pm1.20}$ | $46.86_{\pm1.14}$ | $47.80_{\pm1.76}$ | $64.13_{\pm1.68}$ |
| | min pooling | $72.29_{\pm1.06}$ | $47.08_{\pm0.91}$ | $48.80_{\pm1.42}$ | $64.60_{\pm1.93}$ |
| | max pooling | $72.47_{\pm0.88}$ | $47.10_{\pm0.82}$ | $49.02_{\pm1.13}$ | $64.23_{\pm1.70}$ |
| | Uni-DHC | $\mathbf{72.67_{\pm0.99}}$ | $\mathbf{47.44_{\pm0.99}}$ | $\mathbf{49.31_{\pm1.45}}$ | $\mathbf{65.42_{\pm1.87}}$ |
| Actor | w/o SE | $67.18_{\pm1.20}$ | $18.14_{\pm2.44}$ | $29.75_{\pm7.04}$ | $52.05_{\pm2.38}$ |
| | min pooling | $60.54_{\pm3.47}$ | $16.80_{\pm5.88}$ | $20.42_{\pm4.92}$ | $51.88_{\pm4.70}$ |
| | max pooling | $60.02_{\pm3.33}$ | $16.14_{\pm6.51}$ | $17.62_{\pm6.20}$ | $49.59_{\pm6.52}$ |
| | Uni-DHC | $\mathbf{67.57_{\pm1.44}}$ | $\mathbf{18.30_{\pm2.94}}$ | $\mathbf{31.30_{\pm5.56}}$ | $\mathbf{53.60_{\pm4.09}}$ |
| Cornell | w/o SE | $57.12_{\pm2.39}$ | $34.16_{\pm3.86}$ | $27.92_{\pm2.34}$ | $43.18_{\pm3.82}$ |
| | min pooling | $51.42_{\pm2.18}$ | $25.32_{\pm3.70}$ | $17.30_{\pm2.89}$ | $40.31_{\pm2.03}$ |
| | max pooling | $51.31_{\pm1.53}$ | $24.23_{\pm2.97}$ | $16.43_{\pm2.65}$ | $39.72_{\pm2.57}$ |
| | Uni-DHC | $\mathbf{58.03_{\pm2.41}}$ | $\mathbf{35.06_{\pm3.91}}$ | $\mathbf{28.83_{\pm2.52}}$ | $\mathbf{44.24_{\pm3.84}}$ |

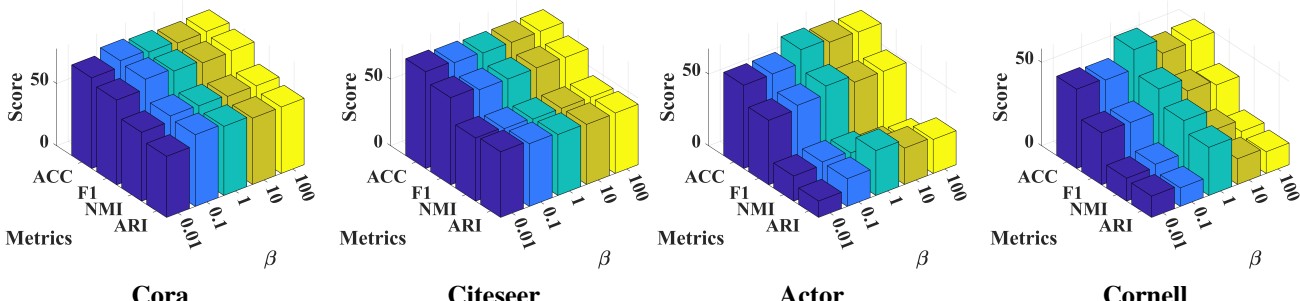

*Figure 7.* The sensitivity analysis of hyper-parameter $\beta$ across four evaluation metrics on four datasets.

