# OpenReview forum: "A Unified Framework for Deep Hypergraph Clustering Beyond Homophily"
_ICML.cc/2026/Conference — ICML 2026 regular_

### Official Review · Reviewer_HsHs · 2026-03-01

**Soundness:** 2
**Presentation:** 3
**Significance:** 3
**Originality:** 2
**Overall Recommendation:** 3
**Confidence:** 4

**Summary:**

This paper addresses hypergraph clustering under heterophily, where connected nodes may not be similar. While most existing work rely on fixed propagation under homophily assumptions, the authors present Uni-DHC that uses learnable hypergraph propagation with cross-view alignment and hyperedge-level correlation reduction. Experimental results show that Uni-DHC consistently outperforms baselines, especially on heterophilic datasets.

**Compliance With Llm Reviewing Policy:**

Affirmed.

**Final Justification:**

I appreciate the authors for their response. They partially addressed my concerns, and I have updated my score accordingly. However, some issues remain:

- **Homophily observation:** While the authors clarified the definition and plan to revise Figure 1, it remains unclear how to rigorously distinguish homophily vs. heterophily using absolute scalar values. In particular, without a proper reference (e.g., random or null models), the interpretation is less convincing. As the authors mentioned, I suggest to further strengthen this part in the final version and better position their measure relative to existing definitions in prior work.
- **Method positioning:** I appreciate the authors’ plan to revise the framing in the final version. However, the proposed method appears to be a general unsupervised representation learning approach rather than one specifically tailored to clustering. In that case, it would be beneficial to provide broader empirical validation beyond clustering (e.g., on tasks such as node classification or link prediction).

Overall, the rebuttal improved clarity, but these aspects would benefit from further refinement in the final version.

**Key Questions For Authors:**

Please refer to the weaknesses above.

**Limitations:**

Please refer to the weaknesses above.

**Strengths And Weaknesses:**

**Strengths**
- The authors present a simple and principled formulation of multi-order propagation via learnable polynomial filtering.
- The method is empirically effective, especially on heterophilic datasets.
- The paper is overall well-written and easy to follow.

**Weaknesses**
- I have some concerns regarding Figure 1:
  - Homophily ratio is not formally defined.
  - The figure reports only absolute homophily ratios, and thus it is unclear whether the observed homophily/heterophily is statistically significant.
  - The observation appears inconsistent with prior work such as "VilLain: Self-Supervised Learning on Homogeneous Hypergraphs without Features via Virtual Label Propagation (WWW 2024)" which reports strong multi-order homophily rather than heterophily. The authors discuss or clarify this.
- The model learns node embeddings independently of the clustering objective, and k-means is applied post hoc. Thus, it is unclear why the paper is positioned specifically as a "clustering" framework rather than general unsupervised hypergraph representation learning.
-  The proposed learnable multi-order propagation optimizes global scalar coefficients over predefined propagation steps. Similar layer-wise weighting or polynomial filtering mechanisms have been extensively studied in graph learning. The authors should clearly discuss what is their technical novelty compared to these:
   - Adaptive Universal Generlaized Pagerank Graph Neural Network (ICLR 2021)
   - MixHop: Higher-Order Graph Convolutional Architectures via Sparsified Neighborhood Mixup (ICML 2019)
- It would be interesting to analyze the learned layer-wise coefficients after training.

---

> ### Author Rebuttal · Authors · 2026-03-31
>
> Thank you very much for your insightful question and valuable comments. Here are our responses:
>
> **1. About Fig. 1: unclear homophily definition, lack of statistical significance, and inconsistency with prior work.**
>
> **Response:** First, regarding the formal definition, the homophily ratio used in Figure 1 is already defined in the Appendix. Second, we examined one-hop and two-hop homophily on additional heterophilic hypergraph datasets (Table 1) and observed the same trend: homophily varies across propagation ranges. In particular, two-hop homophily is consistently higher than one-hop homophily across all datasets. This suggests statistical significance shown in Fig. 1. We will revise Fig. 1 according to the comment. Third, our observation is not intended to contradict VilLain, since the two works quantify different notions of structural consistency under different settings. Specifically, VilLain measures hyperedge-level label homogeneity by examining the entropy of label distributions within hyperedges and studies homogeneous hypergraphs without features. In contrast, Fig. 1 reports node-level homophily under one- and two-hop propagation, defined as the ratio of same-label weighted connections after normalized clique expansion. These notions are not directly comparable, especially in heterogeneous or mixed-label settings.
>
> | Dataset | One-hop Homophily | Two-hop Homophily |
> |:--------|:------------------:|:------------------:|
> | Pokec   | 0.45              | 0.52              |
> | Actor   | 0.46              | 0.49              |
> | House   | 0.49              | 0.50              |
> | Senate  | 0.47              | 0.50              |
>
> **2. Why is this framed as clustering rather than general unsupervised representation learning?**
>
> **Response:** We position it as a clustering framework because our downstream tasks for evaluation focus on node clustering. We will clarify it in following the suggestion.
>
> **3. Technical novelty compared with GPR-GNN and MixHop.**
>
> **Response:** We agree that learning or mixing multi-order propagation has been studied in graph learning, but it has been rarely studied in hypergraph learning fields. The main differences are: **1. Different structural setting.** GPR-GNN and MixHop are developed for pairwise graphs, whereas our method is built on a hypergraph-induced propagation operator that explicitly models node–hyperedge–node high-order relations. Therefore, our propagation mechanism is designed for hypergraph structures rather than directly inherited from graph-based propagation. **2. Different downstream tasks and optimization objectives.** GPR-GNN and MixHop are mainly proposed as propagation mechanisms for supervised or semi-supervised graph learning, while our method is developed for unsupervised hypergraph clustering and is jointly optimized with node-level cross-view alignment and hyperedge-level correlation reduction. **3. Different theoretical contribution.** More importantly, we provide a spectral interpretation showing that conventional HGNN-style propagation corresponds to a fixed low-pass filter, while our learnable multi-order propagation induces a data-adaptive polynomial spectral filter. This theoretical perspective helps explain why adaptive propagation is beneficial beyond homophily: in heterophilic hypergraphs, useful information may not be concentrated only in low-frequency components, and fixed smoothing may therefore be suboptimal.
>
> **4. Analysis of learned coefficients.**
>
> **Response:** We additionally analyzed the learned coefficients $\gamma_k$ after training on several datasets. We observe that the $\gamma_k$ exhibits clear dataset-dependent patterns rather than a fixed or monotonic trend. As shown in Table 2, on homophilic datasets (Cora, Citeseer), the $\gamma_k$ distributes more smoothly across multiple propagation orders, indicating that information from low- to mid-order neighbors can be jointly useful. In contrast, on heterophilic datasets such as Actor and Cornell, the model places much stronger emphasis on $k=0$ together with a few selective propagation orders, while some higher-order coefficients are strongly suppressed and can even become negative. This behavior is consistent with our spectral interpretation: the learned coefficients act as a data-adaptive polynomial filter rather than a predefined smoothing scheme. These notions are different and not directly comparable, especially in mixed-label settings.
>
> | Dataset  | k=0 | k=1 | k=2 | k=3 | k=4 | k=5 | k=6 | k=7 | k=8 | k=9 | k=10 |
> |:----------|:---:|:---:|:---:|:---:|:---:|:---:|:---:|:---:|:---:|:---:|:----:|
> | Cora     | 0.41 | 0.31 | 0.35 | 0.35 | 0.32 | 0.30 | 0.28 | 0.26 | 0.24 | 0.18 | 0.14 |
> | Citeseer | 0.34 | 0.22 | 0.26 | 0.20 | 0.18 | 0.17 | 0.19 | 0.21 | 0.22 | 0.19 | 0.16 |
> | Actor    | 0.95 | 0.08 | 0.35 | 0.01 | -0.02 | -0.10 | -0.06 | -0.03 | 0.01 | 0.05 | 0.09 |
> | Cornell  | 0.88 | -0.12 | 0.36 | 0.14 | 0.12 | 0.08 | 0.05 | 0.03 | 0.02 | 0.01 | 0.01 |

---

> > ### Author Rebuttal · Reviewer_HsHs · 2026-03-31
> >
> > Thank you for the response. While the rebuttal clarified some of my concerns, several core issues remain insufficiently addressed.
> >
> > Specifically, my concerns regarding (3) and (4) are addressed. Thank you for the clarification.
> >
> > However, concerns (1) and (2) remain.
> >
> > Regarding (1),
> > - Although the definition is provided in Appendix, Figure 1 serves as a key motivation and should be self-contained.
> > - Reporting absolute homophily values is insufficient to determin whether a dataset is truly homophilic or heterophilic. Such values require a meaningful baseline or reference (e.g., random null models) for proper interpretation.
> > - Since homophily in hypergraphs has been studied in prior work such as VilLain, I suggest discussing the similarities and differences in the main text. For example, the authors may consider positioning their definition as complementary to existing measures.
> >
> > Regarding (2),
> > - I believe that evaluating on clustering task does not sufficiently jsutify for framing the method as a "clustering" framework. The proposed objective focuses on general representation learning, and clustering is applied post hoc. Without a clustering-specific objective or joint optimization, this is more appropriately viewed as unsupervised representation learning.

---

> > > ### Author Response · Authors · 2026-04-02
> > >
> > > **Response to (1): Clarification of homophily definition and Fig. 1.**
> > >
> > > Thank you for the insightful comment.
> > >
> > > **1. Self-contained definition in Fig.1.**  We will move the formal definition of the homophily ratio to the main text to ensure that Fig. 1 is self-contained and logically complete.
> > >
> > > **2. Interpretation of absolute homophily values.**  Following common practice in the graph learning literature [1][2], we use the homophily ratio as an empirical indicator of structural regimes, where values above 0.5 are generally associated with homophilic graphs and values below 0.5 with heterophilic ones. Nevertheless, we will follow the suggestion from the reviewer and adjust the definition of the homophily rate to provide a more fine-grained description for it. Specifically, we will incorporate reference baselines (e.g., random-label baselines) to provide a more meaningful context for interpreting the reported values.
> > >
> > > **3. Relation to prior work (e.g., VilLain).**  We appreciate the suggestion regarding prior work such as VilLain. Our definition measures **node-level homophily** under different propagation ranges, while VilLain focuses on **hyperedge-level label homogeneity** via entropy in featureless settings. We will explicitly clarify in the main text that the two measurements are complementary rather than contradictory, and discuss their differences in terms of granularity and application scenarios.
> > >
> > > [1] Zheng Z, Bei Y, Zhou Z, et al. Understanding and Enhancing Message Passing on Heterophilic Graphs via Compatibility Matrix[C]//The Thirty-ninth Annual Conference on Neural Information Processing Systems, 2025.
> > >
> > > [2] Shen Z, Kang Z. When heterophily meets heterogeneous graphs: Latent graphs guided unsupervised representation learning[J]. IEEE Transactions on Neural Networks and Learning Systems, 2025.
> > >
> > > **Response to (2): On the positioning as a clustering framework.**
> > >
> > > Thank you for this important observation. Our method can be more appropriately viewed as an unsupervised representation learning framework for clustering.
> > > Following this suggestion, we will revise the paper to clarify this positioning and avoid potential ambiguity. Specifically, we will describe our method as an unsupervised representation learning framework for hypergraphs. At the same time, we emphasize that our method is applied to clustering tasks and we designed a clustering-oriented objective, i.e., the hyperedge-level decorrelation loss, which explicitly reduces redundancy across groups and helps improve inter-cluster separability. Thus, we evaluate the learned representations using standard clustering methods (e.g., k-means). We will make this distinction explicit in the revised version.
> > >
> > > We hope this clarification addresses your concerns. Please let us know if any further clarification would be helpful.

---

### Official Review · Reviewer_Sc13 · 2026-03-07

**Soundness:** 4
**Presentation:** 3
**Significance:** 4
**Originality:** 4
**Overall Recommendation:** 5
**Confidence:** 5

**Summary:**

The paper discusses the heterophilic issue in hypergraph clustering and develops a novel and unified framework for deep hypergraph clustering method. The developed framework includes three key components, i.e., the learnable high-order hypergraph propagation mechanism, the node-level cross-view alignment mechanism, and the hyperedge-level correlation reduction mechanism. Apart from the designed method, another contribution is that the work analyzes the homophily at different ranges. To validate the effectiveness and superiority, the work conducts a number of theoretical analyses and compares the method with several state-of-the-art methods.

**Compliance With Llm Reviewing Policy:**

Affirmed.

**Final Justification:**

The authors propose a novel method with a clear motivation,  and the overall method design is technically sound. After reading the rebuttal, I find all my concerns adequately addressed through more explanations and deep analysis. Therefore, I raise my score from weak accept to accept.

**Key Questions For Authors:**

1. In Appendix D, the authors construct heterophilic hypergraphs from graph benchmarks by forming hyperedges from each node and its one-hop neighbors. How does it ensure that the construction method preserves the heterophilic nature of the original graph? Could this introduce artificial homophily?

2. The SE-based channel recalibration is mentioned in Algorithm 1 but not deeply analyzed. Could the authors provide more insight into its contribution?

3. For the hyperedge-level loss in Eq. (9), you use mean pooling to obtain hyperedge representations. How sensitive are the results to the choice of pooling function? Have you compared with sum or max pooling?

**Limitations:**

Refer to the Weaknesses and the Questions.

**Strengths And Weaknesses:**

The strengths of the paper are:

1. A novel hypergraph clustering method.

2. Effectiveness on heterophilic datasets.

3. Clear theoretical interpretation.

The weaknesses of the work are:

1. The construction of heterophilic hypergraphs from graph datasets is interesting but may introduce bias.

2. The role of the SE-based recalibration module is not deeply analyzed.

---

> ### Author Rebuttal · Authors · 2026-03-31
>
> Thank you very much for your insightful question and valuable comments.
>
> **1.	Potential bias in constructing heterophilic hypergraphs.**
>
> **Response:** We appreciate this important concern. In Appendix D, we construct hyperedges by grouping each node with its one-hop neighbors, which is designed to preserve the original local structure rather than alter it. Importantly, in heterophilic graphs, a node’s one-hop neighbors are predominantly from different classes, and thus forming hyperedges in this way naturally results in heterogeneous (mixed-label) hyperedges that preserve the heterophilic nature of the original graph. Regarding the concern of introducing artificial homophily, we emphasize that we do not modify or filter edges but directly use the original adjacency structure, no label information is involved in hyperedge construction, and the process only reorganizes pairwise relations into higher-order groups without changing the underlying connectivity. Therefore, the resulting hypergraph inherits the structural properties of the original graph and, in fact, makes the heterophilic nature more explicit at the group level rather than introducing homophily. We will clarify this point and include additional discussion in the revised version.
>
> **2. Lack of analysis of the SE-based recalibration module.**
>
> **Response:** Thank you for pointing this out. The SE-based channel recalibration module plays an important role in our framework, and we will provide a clearer explanation in the revision. Specifically, the SE module is applied before high-order propagation, and its purpose is to adaptively reweight feature channels based on global statistics. Given input features $X \in \mathbb{R}^{n\times d}$, the module computes channel-wise importance via global pooling and a gating mechanism, producing recalibrated features $X’$. Its contribution can be understood from two aspects: **1. Suppressing noisy or irrelevant feature channels.** In hypergraph settings, especially under heterophily, propagation may amplify noisy signals from structurally inconsistent neighbors. The SE module mitigates this by down-weighting less informative channels before propagation. **2. Enhancing informative signals for propagation.** By emphasizing discriminative feature dimensions, the recalibrated features provide a more reliable input for subsequent multi-order propagation, leading to improved representation quality. **3. Complementarity with learnable propagation.** While the propagation module adaptively controls the range of information aggregation (structural aspect), the SE module operates in the feature space (attribute aspect). Together, they enable joint adaptation over both structure and features. We also note that this module is lightweight and introduces negligible computational overhead. Empirically, we observe consistent performance improvements when including SE-based recalibration. We will include additional analysis and ablation results to better demonstrate its contribution in the revised manuscript.
>
> **3. How sensitive is performance to the choice of pooling function?**
>
> **Response:**  In Eq. (7), we adopt mean pooling by default to construct hyperedge representations, since it naturally normalizes for hyperedge size and yields more balanced group-level representations for the hyperedge-level correlation reduction objective. We have compared mean pooling with min pooling and max pooling in Appendix E. The results show that average pooling yields more stable and competitive performance overall, especially on heterophilic datasets such as Actor and Cornell, while min/max pooling tend to amplify noisy or uninformative node features within hyperedges and lead to inferior clustering results. We did not include sum pooling in the current submission. Following the reviewer’s suggestion, we will additionally evaluate sum pooling and report the corresponding results in the revised version.

---

> > ### Author Rebuttal · Reviewer_Sc13 · 2026-04-03
> >
> > The authors have provided sufficient clarifications to address my concerns. Specifically, the authors explain the potential bias issue in constructing heterophilic hypergraphs and provide a deep analysis of the SE-based channel recalibration as well as the polling strategies. The explanations are clear and reasonable, and I support its acceptance.

---

### Official Review · Reviewer_gSe4 · 2026-03-11

**Soundness:** 4
**Presentation:** 3
**Significance:** 3
**Originality:** 4
**Overall Recommendation:** 5
**Confidence:** 4

**Summary:**

This work presents a deep hypergraph clustering method named Uni-DHC that addresses heterophily via learnable high-order propagation and dual self-supervised learning. Besides, it analyzes the propagation mechanism and its relationship with adaptive polynomial filtering from spectral perspective. The authors evaluate the proposed Uni-DHC on eight datasets.

**Compliance With Llm Reviewing Policy:**

Affirmed.

**Final Justification:**

After reading the response and other comments, all my concerns have been addressed. I will maintain my positive score.

**Key Questions For Authors:**

Q1. The experiments report that the work is more efficient than the compared works, but which modules of the method contribute most to the efficiency?

Q2. The paper focuses on unsupervised clustering. Could the proposed framework be extended to supervised or semi-supervised learning tasks, such as node classification? If so, what modifications would be needed?

**Limitations:**

The authors should analyze its application scenarios to demonstrate its significance.

**Strengths And Weaknesses:**

**Strengths**

S1. This work is effective for handling of heterophily.

S2. The paper is with clear motivations and detailed introduction to the framework.

S3. The proposed framework is with impressive empirical results.

**Weaknesses**

W1. Lacks analysis of computational cost and scalability with increasing hypergraph order or dataset size.

W2. Lacks systematic comparison with heterophily-specific baselines across datasets with varying heterophily ratios.

---

> ### Author Rebuttal · Authors · 2026-03-31
>
> Thank you very much for your insightful question and valuable comments.
>
> **1. Lacks analysis of computational cost and scalability.**
>
> **Response:** In addition to the empirical efficiency comparison (Table 4), we further clarify the scalability of Uni-DHC with respect to both hypergraph order and dataset size. The dominant cost lies in the learnable high-order propagation in Eq.(3), which scales approximately as O(K|E|d), where K is the propagation order and |E| denotes the number of nonzero hypergraph entries. This linear dependency ensures good scalability to large and sparse hypergraphs. Importantly, unlike HGNN-style methods that stack multiple hypergraph convolution layers, our formulation uses a weighted combination of multi-order propagation, avoiding repeated expensive message passing. Therefore, increasing the propagation order K introduces only linear overhead rather than deep stacking costs. With respect to dataset size, the use of sparse structures and lightweight MLP encoders enables the model to scale efficiently with the number of nodes. We will include a more explicit discussion of scalability in the revised version.
>
> **2. Lacks comparison with heterophily-specific baselines across varying heterophily levels.**
>
> **Response:** Our current evaluation includes both homophilic and heterophilic datasets (e.g., Actor, Cornell, Texas, and Wisc), which already cover a range of heterophily levels. The results consistently demonstrate the effectiveness of Uni-DHC in heterophilic settings. We agree that a more systematic comparison across varying heterophily ratios would further strengthen the paper. In the revision, we will (1) explicitly report heterophily statistics of each dataset, and (2) include additional comparisons or discussions with heterophily-oriented methods where applicable. We would like to emphasize that our method is not tailored to a specific heterophily assumption, but instead learns a flexible combination of propagation orders, which naturally adapts to different structural regimes.
>
> **3. Which modules contribute most to efficiency?**
>
> **Response:** The efficiency gain of Uni-DHC mainly comes from three aspects: **1. Learnable high-order propagation without deep stacking.** Instead of stacking multiple hypergraph convolution layers, our model computes a weighted combination of multi-order propagation in a single module, which significantly reduces repeated message passing. **2. Lightweight MLP-based encoding.** After propagation, representation learning is performed by simple MLPs rather than heavy hypergraph convolutions, leading to lower computational overhead. **3. Efficient objective design.** The node-level alignment only involves diagonal entries, avoiding full pairwise supervision, and the hyperedge-level decorrelation is computed once per iteration. This avoids the expensive negative sampling and pairwise computations required in contrastive learning. These components jointly contribute to the favorable efficiency-performance trade-off observed in Table 4. We will clarify this in the revised manuscript.
>
> **4. Can the framework be extended to supervised/semi-supervised tasks?**
>
> **Response:** Yes, the proposed framework can be naturally extended to supervised or semi-supervised tasks such as node classification. Specifically, the learnable high-order propagation module can be directly used as a feature transformation backbone. To adapt the framework: **1. Supervised setting.** A classification head (e.g., linear layer + softmax) can be added on top of the learned representations, and the objective can be replaced with a standard cross-entropy loss. **2. Semi-supervised setting.** The current unsupervised objectives (node-level alignment and hyperedge-level decorrelation)can be retained as auxiliary regularizers, while combining them with a supervised loss on labeled nodes.Such an extension would preserve the key advantage of our framework—-adaptive multi-order propagation—-while enabling it to benefit from label information. We will include a discussion of this extension in the revised version.

---

> > ### Author Rebuttal · Reviewer_gSe4 · 2026-04-02
> >
> > Thanks for the response. All my concerns have been addressed. I will maitain my score.

---

### Official Review · Reviewer_MeHM · 2026-03-13

**Soundness:** 3
**Presentation:** 3
**Significance:** 4
**Originality:** 4
**Overall Recommendation:** 5
**Confidence:** 5

**Summary:**

The paper introduces Uni-DHC, a deep hypergraph clustering framework designed to handle heterophily in hypergraph clustering. It combines learnable high-order propagation with two self-supervised objectives: node-level cross-view alignment and hyperedge-level correlation reduction. The method is evaluated on both homophilic and heterophilic benchmarks, demonstrating its superior performance.

**Compliance With Llm Reviewing Policy:**

Affirmed.

**Final Justification:**

The author answered my questions about this article, and I am satisfied with the quality of the article and the discussion.

**Key Questions For Authors:**

1. For the learnable coefficients controlling the propagation orders, how are these weights initialized at the beginning of training?
2. The node-level cross-view alignment uses MSE loss to enforce diagonal entries to 1. However, there might be some other methods like contrastive learning, which I think could better separate nodes.
3. The node-level loss in Eq. (6) only considers the diagonal entries of $S^{v}$ by enforcing them to be close to 1, but leaves the remaining entries unconstrained. Please explain the justifications for this design.

**Limitations:**

The paper presents a novel method about hypergraph clustering, but some design lacks full explanations and might be further improved.

**Strengths And Weaknesses:**

Strengths:
1. The paper presents a novel unified framework with clear modular design.
2. The paper demonstrates superior results on heterophilic datasets.
3. The paper demonstrates efficient training due to unshared MLPs and learnable propagation.

Weakness:
1. The trade-off parameter $\beta$ is not deeply analyzed in the main text.
2. The paper could benefit from a discussion on computational complexity beyond training time.

---

> ### Author Rebuttal · Authors · 2026-03-31
>
> Thank you very much for your valuable comments.
>
> **1. $\beta$ is not deeply analyzed**
>
> **Response:** In the current submission, due to space limitations, we reported the sensitivity analysis of $\beta$ in the appendix, while the main text only briefly mentioned it. We will make the discussion more explicit in the revised version.
>
> **2. Lack of computational complexity discussion**
>
> **Response:** We thank the reviewer for pointing this out. In addition to the empirical training time comparison (Table 4), we further analyze the computational complexity of Uni-DHC. The main cost comes from the learnable high-order propagation in Eq.(3), which involves $K$-step propagation and scales approximately as $\mathcal{O}(K|E|d)$, where $|E|$ denotes the number of nonzero hypergraph entries and $d$ is the feature dimension. The subsequent MLP encoders introduce a standard cost of $\mathcal{O}(nd^2)$, which is lightweight compared to repeated hypergraph convolutions. For the node-level alignment, although a similarity matrix is defined, the loss only depends on diagonal entries and can be computed without explicitly forming the full $n\times n$ matrix. The hyperedge-level correlation reduction involves aggregation with cost $\mathcal{O}(|E|d)$ and correlation computation at the hyperedge level, which is performed once per iteration. Overall, Uni-DHC avoids repeated expensive hypergraph convolution operations and instead relies on efficient propagation and lightweight MLPs, achieving a favorable balance between efficiency and performance, as also reflected in Table 4.
>
> **3. How are the propagation coefficients initialized?**
>
> **Response:** In Eq.(3), the learnable coefficients $\(\gamma_k\)^{K}_{k=0}$ are used to weight different propagation orders, and in our implementation they are initialized uniformly as $\gamma_k=\frac{1}{K+1}$ for all $k \in \{0, …, K\}$. This initialization ensures that all propagation orders contribute equally at the beginning of training, providing a neutral starting point without bias toward any specific neighborhood range.
>
> **4. Why use MSE-based alignment instead of contrastive learning?**
>
> **Response:** While contrastive learning is effective in many representation learning tasks, our choice of an MSE-based alignment objective is intentional. The key reason is that, in unsupervised hypergraph clustering, especially under heterophily, explicitly treating non-matching nodes as negatives may introduce false negatives, since nodes that are not immediate counterparts across views may still belong to the same latent cluster or share meaningful semantics, and forcing them apart could harm clustering quality. Our node-level objective is therefore designed to emphasize cross-view consistency of the same node rather than explicit pairwise repulsion among different nodes; specifically, Eq.(6) encourages corresponding nodes across views to remain aligned, which stabilizes representation learning without relying on negative sampling. In addition, contrastive learning typically requires constructing large numbers of negative pairs and computing pairwise similarities, which introduces non-trivial computational overhead, whereas our MSE-based formulation is more lightweight and efficient. At the same time, discriminability is not ignored in our framework: instead of enforcing separation at the node level, we introduce hyperedge-level correlation reduction (Eq.(9)), which explicitly suppresses redundancy among hyperedge representations. This division of roles is deliberate—node-level alignment improves stability, while hyperedge-level decorrelation enhances diversity and discriminability. We will make this motivation clearer in the revised version.
>
> **5. Why does the node-level loss only constrain diagonal entries?**
>
> **Response:**  In Eq.(6), the node-level loss only enforces the diagonal entries of the similarity matrix $S^v$ to approach 1 while leaving off-diagonal entries unconstrained. This design is motivated by several considerations. First, in unsupervised clustering, the true relationships between nodes are unknown, and enforcing off-diagonal entries to be small (as in typical contrastive objectives) may incorrectly push latent positive pairs apart. Second, by only aligning the diagonal entries, we ensure cross-view consistency of the same node while preserving flexibility in the representation space, allowing the model to organize inter-node relationships according to the data distribution. Third, redundancy reduction is explicitly handled at the hyperedge level via Eq.(9), which penalizes off-diagonal correlations and complements the node-level objective. This leads to a clear division of roles, where node-level alignment improves consistency and stability, while hyperedge-level decorrelation enhances diversity and discriminability. Overall, this design avoids over-constraining the model and is particularly beneficial in heterophilic settings, where strict node-level separation may be misleading.

---

> > ### Author Rebuttal · Reviewer_MeHM · 2026-04-02
> >
> > The author answered my questions.

---

### Decision · Program_Chairs · 2026-04-30

**Decision:**

Accept (regular)

**Comment:**

This paper proposes a unified framework for deep hypergraph clustering, which deals with heterophily through learnable higher-order propagation. Extensive experiments demonstrate its superiority. However, the reviewers indicated that there are still some unresolved issues, such as the lack of reference baselines for homophily characterization. Moreover, this method studies clustering problems, but the proposed method is a general representation learning problem and lacks cluster-specific components for effective cluster analysis and generalization specification. At the same time, there are still concerns about the novelty of higher order propagation methods regarding this method. In general, the proposed framework needs to be further refined and enriched.

The authors have received a lot of actionable feedback to improve the paper in the reviews, and they should use it to improve the final version.